# Social-Emotional Development of Children in Asia: A Systematic Review

**DOI:** 10.3390/bs13020123

**Published:** 2023-02-01

**Authors:** Geok Har Yong, Mei-Hua Lin, Teck-Hock Toh, Nigel V. Marsh

**Affiliations:** 1Department of Psychology, School of Medical and Life Sciences, Sunway University, Bandar Sunway 47500, Malaysia; 2Department of Paediatrics & Clinical Research Centre, Sibu Hospital, Ministry of Health Malaysia, Sibu 96000, Malaysia; 3School of Social and Health Sciences, James Cook University, Singapore 387380, Singapore

**Keywords:** social-emotional development, children, Asia, culture, social competence, emotional development

## Abstract

There has been growing interest in the social-emotional development of children. However, the social-emotional development of children in Asia remains a knowledge gap. This systematic review identifies and summarizes existing studies on children’s social-emotional development in Asia. We conducted a systematic review using the Guidelines for Preferred Reporting Items for Systematic Reviews and Meta-analyses (PRISMA). We reviewed 45 studies that met the inclusion criteria, and they were from 12 Asian countries, primarily the East Asia region (China and Hong Kong). Most of the studies were cross-sectional in design (*n* = 28, 62.2%). Six themes emerged, including (a) social-emotional development (overall) (*n* = 24, 53.3%); (b) social competence (*n* = 7, 15.6%); (c) emotional development (*n* = 5, 11.1%); (d) social-emotional learning (*n* = 3, 6.7%); (e) problem behavior (*n* = 3, 6.7%); (f) self-regulation (*n* = 2, 4.4%); and (g) both social-emotional learning and problem behavior (*n* = 1, 2.2%). The findings highlighted the paucity of studies, the need for examining more diverse variables in a similar population, and the low quality of intervention studies in social-emotional research in Asia. Research gaps indicate the need for more social-emotional and ethnocultural studies in other Asian regions. Parent and teacher knowledge of children’s social-emotional functioning should be examined more closely in future research.

## 1. Introduction

Social-emotional development is a process where an individual acquires and applies the knowledge, skills, and attitudes to develop healthy identities, manage emotions, achieve personal goals, show and feel empathy, build positive relationships, and make responsible decisions [1]. The social-emotional domain is an important developmental milestone in early childhood [2]. Bronfenbrenner’s social-ecological model illustrated that the family environment closest to the child might influence the child’s development status, including social-emotional development [3]. Bronfenbrenner’s model has proposed five levels, known as systems, that can impact the child’s development; the microsystem, mesosystem, exosystem, macrosystem, and chronosystem. These five systems are interrelated and influence the child’s development [4]. The microsystem, the first level of the system, includes influences such as family, school, peers, neighborhood, health services, and religious entities that directly impact the child’s development. Within this system, the child has a bidirectional relationship with their environment. The child’s immediate environment can influence how the child learns, reacts, and behaves toward the environment and people [4,5]. A positive or negative home environment might influence the child’s social-emotional development. Parental perception of social-emotional development may directly influence the child’s development. The child’s social-emotional development may differ based on their parent’s norms or culture. Thus, it is crucial to understand the child’s social-emotional development within a cultural context.

Many studies have also shown that social-emotional development is associated with children’s success in school, academic grades, and adulthood [2,3,6,7]. Recent studies have reported that social-emotional competence is a strong predictor of school readiness, academic achievements, and the psychological well-being of a child [8,9,10]. Children high in social-emotional competence were reported to have the ability to form positive relationships with others and regulate and express emotions in a culturally appropriate way, and they displayed assertive self-regulatory behavior that led to success in school, work, and daily life [11]. In contrast, children with low social-emotional competence experience delinquency issues, academic failure, behavioral problems in school, and substance use and abuse in later life [12]. Furthermore, poor social-emotional competence is also associated with poorer physical health, higher risk of financial struggles, more mental health issues, and increased criminal offenses in adulthood [6,13].

Cultural differences in social-emotional development are a vital aspect to consider in early childhood development research. Appropriate social behavior and emotional expressivity in a particular country depends largely on that country’s cultural norms and values [14]. For example, emotions and feelings are rarely explicitly communicated in public among East Asian culture. Group harmony and relatedness, with the associated moderation in all matters of the heart, is one of the core values within the East Asian culture [15]. However, in another culture, the cultivation of emotional expression may be highly encouraged as it is viewed positively, representing individuality, autonomy, and truth to self.

Similarly, parental beliefs and parenting practices are also shaped by cultural values and norms [16]. Most East Asian cultures, particularly Chinese, often use authoritarian parenting, where parents are strict, have high control, and show low warmth toward their children. This type of parenting within East Asian cultures is heavily influenced by Confucianism teaching that focuses on filial piety, the lifelong obligation to the family, and interpersonal harmony [17]. From young, children are taught to have self-control, be obedient to their parents, respect the elderly, and be modest [18]. In contrast, Western cultures often use authoritative parenting, where the parent is not as controlling and expresses more warmth toward their children [17]. Many studies have reported that parenting styles influence children’s development, particularly social-emotional development [17,19,20]. Children with authoritarian parenting are often reported to have higher internalizing problems, poor social competence, and maladaptive development, while children with authoritative parenting exhibit independence and better social competence. It should be noted, however, that the definitions of social-emotional competence are derived from Western cultures [21]. In contrast, based on the Western cultures’ definitions of social competence, children that experienced authoritarian parenting may have maladaptive social-emotional development within the East Asian culture, being quiet, shy, and reserved in front of authority, such as teachers and the elderly, and obeying instructions are perceived as socially well-mannered. Other international studies have reported cultural differences between Eastern and Western cultures. Thus, it is vital to understand and interpret children’s emotions and behavior within the context of the children’s culture itself [17,22].

Social-emotional behavior may be valued and perceived differently across different contexts and cultural groups [23,24]. However, research on social-emotional development is mainly from North America, Europe, the United Kingdom, and Australia [25,26]. As the importance of social-emotional development in children has become more widely acknowledged by parents, teachers, researchers, and policymakers outside of Asia, studies on social-emotional development have begun to gain attention in Asia, in terms of both the impact of cultural differences and the social-emotional development itself [27]. Although there are studies published on children’s social-emotional development in Asia, these have not been systematically summarized. A synthesis of these studies will help identify and map the findings of these studies of social-emotional development in Asia, thereby providing a more precise understanding of the research findings to date.

This review aims to explore and summarize the studies of social-emotional development conducted in Asian countries. This review answers the questions: (1) What are the Asian countries that have conducted studies on social-emotional development in children? (2) What are the types of study designs used to examine the social-emotional development of children in Asia? (3) What domains of social-emotional development have been studied in Asia? (4) What are their key findings? This review aims to summarize the existing peer-reviewed social-emotional research conducted in “Asia” and provide suggestions for future research.

## 2. Materials and Methods

This review utilized a systematic review process to explore the social-emotional development research conducted with children in Asia. We conducted this systematic review based on the Guidelines for Preferred Reporting Items for Systematic Reviews and Meta-analyses (PRISMA) [28]. The protocol for this review was registered in PROSPERO (CRD42021238826). The systematic review involved the four stages below:

### 2.1. Stage 1: Identify the Search Strategy

An electronic search was conducted by GHY and JX in four electronic databases and one web search engine interface, including PubMed, ScienceDirect, Ebscohost, and Google Scholar, on 4 January 2021, using the following search terms:

(“social-emotional” OR “socioemotional” OR “emotional intelligence” OR “social intelligence”) AND (“children” OR “toddler”) AND (“Asia”). The complete search strategies are provided in Appendix A.

Besides the databases search, the reviewers further searched the reference lists of retrieved studies for other potentially relevant studies. The reviewers (GHY and JX) only included primary research articles that were in line with the aims of this review. The reviewers did not restrict the years of the publication to minimize introduction to bias. The reviewers screened the databases for eligible studies based on titles and abstracts. Following that, the reviewers independently screened for full texts for potentially eligible studies. The reviewers used the Mendeley software to verify the existence of duplicate references. The selection procedure was documented according to PRISMA and reported in a flowchart (see Figure 1). The reviewers (GHY and JX) contacted the authors if the relevant articles were not available in full-text. All discrepancies among reviewers (GHY and JX) were resolved through discussion and guidance from the third reviewer (MHL).

### 2.2. Stage 2: Inclusion and Exclusion Criteria

The inclusion criteria for this systematic review were: (1) original studies (both observational and experimental); (2) study participants aged ≤18 years; (3) social-emotional functioning must be either the main focus of the study or one of the main domains studied; (4) studies conducted within Asia; and (5) this systematic review study will also include studies related to social-emotional constructs (e.g., self-regulation). The exclusion criteria were: (1) dissertations, book chapters, unpublished manuscripts, conference abstracts, and review articles; and (2) non-English articles. The definition of children varies across Asia. Thus, this review study defined children as any person aged 0–18 years old. This definition is consistent with the UNICEF Convention on the Rights of the Child definition [30]. Asia is the world’s largest and most populous continent with about 60% of the world’s population [31]. The Asia continent consists of six main geographical regions: Northern Asia, Western Asia, Central Asia, Eastern Asia, Southern Asia, and Southeast Asia [32]. This review includes all regions of Asia to reduce the definition bias of “Asia”.

### 2.3. Stage 3: Data Extraction and Quality Assessment

The reviewers conducted the following steps to decide on the inclusion studies: (1) search the four databases and web search engine using the search strategy mentioned above; (2) remove duplication and merge the search results using Mendeley software; (3) remove non-eligible studies using titles and abstract; (4) retrieve and examine eligible studies for full-text potential; (5) apply the inclusion criteria and shortlist the studies; (6) discussion and resolution between the reviewers on study inclusion and exclusion.

All data were independently extracted by the reviewers (GHY and JX) and entered into Microsoft Excel. Data extracted included the characteristics of the studies (author, year of publication, country of the study, aims of the study), the methods used in the studies (age range, disabilities or special population, subjects involved, screening or diagnostic measures, type of study design, social-emotional domains studied), as well as the summary and key findings of the studies. Three corresponding authors were contacted via e-mail to obtain additional information when necessary, and only one author responded. The reviewers (GHY and JX) conducted an inter-rater reliability check based on ten papers to ensure appropriate categorizations, consistency, and credibility of the data. Any disagreements among the reviewers were discussed to reach a consensus. The reviewers (GHY and JX) then independently assessed the studies’ quality using the Strengthening the Reporting of Observational Studies in Epidemiology (STROBE) statement assessment tool [33]. Any discrepancies were resolved through discussion and guidance from the third reviewer (MHL). The STROBE statement assessment tool is a 22-item checklist that should be included in articles reporting for observational research such as cohort, case-control, and cross-sectional studies to ensure adequate reporting and assessment of the strengths and weaknesses of the study. Eighteen checklist items are common to all three observational research designs, and the other four items (i.e., items 6, 12, 14, and 15) are specific variations according to the study design (see Appendix A). The STROBE checklist score is a summation of the total number of items applicable, with minimum scores of 23 and maximum scores of 25.

### 2.4. Stage 4: Collating, Summarizing, and Reporting the Results

All study findings were entered into Microsoft Excel. Data extraction and analysis was performed using Microsoft Excel. The reviewers (GHY and JX) were unable to perform a meta-analysis due to the clinical heterogeneity of the study characteristics and findings. Based on the Cochrane Handbook for Systematic Reviews of Interventions, performing meta-analysis with clinically diverse studies can be meaningless and the genuine differences may be obscured [34].

### 2.5. Search Results

The initial search yielded 3377 records in total, with 3360 records found from the four electronic search databases (2992 from Google Scholar, 183 from PubMed, 184 from ScienceDirect, and one from Ebscohost), and 17 studies found by reviewing the reference lists and citation searching. There were 3117 records after the duplicates were removed. Following screening of the titles and abstracts, 117 studies were selected for further evaluation. After further review and application of the selection criteria, 45 studies were selected for inclusion in this review. A PRISMA flow diagram is prepared to illustrate the study selection process (see Figure 1).

The extracted data were summarized by the study’s country, the type of study design, and the social-emotional domains studied. From the extracted data, six themes emerged; which were (1) social-emotional development; (2) social competence; (3) emotional development; (4) problem behavior; (5) self-regulation; and (6) social-emotional learning. One study that could not be categorized as it coded into more than one category. Three studies were collated into one category as they measured the same social-emotional domain but differed in the definition used.

## 3. Results

### 3.1. Summary of the Included Study Characteristics

The majority of the studies were published in 2020 (*n* = 11, 24.4%), followed by seven (15.6%) studies in 2018 and 2019 and six (13.3%) studies in 2017. From 2010 to 2016, one to four (2.2–8.9%) studies were published annually (see Appendix A). A total of 46,625 participants involved in these social-emotional studies were children with typical development, parents and teachers, and 689 were children with disabilities (physical or developmental). A majority of these studies involved only children as an informant (*n* = 18, 40.0%), followed by 17 studies (37.8%) that involved either parent or teacher with the child, five studies (11.1%) that included all three (parent, teacher, child) as informants, three studies (6.7%) that involved teacher only, and two studies (4.4%) involved the parent only. Table 1 presents a summary of the characteristics of the included studies and their key findings.

### 3.2. Summary of Social-Emotional Research in Asian Countries

All 45 studies included for qualitative synthesis were primary studies on social-emotional development published in English and were within the Asia continent. Out of the 45 studies, 13 originated in China, seven in Hong Kong (HK), 3 each in India, South Korea, Malaysia, Singapore, Thailand, and Turkey, and 1 study each in Japan, Pakistan, and Taiwan. Two studies compared social-emotional development in two countries (India and China, U.S. and South Korea) (see Appendix A).

### 3.3. Summary of the Type of Study Design

Among the 45 studies, 28 (62.2%) were cross-sectional, 7 (15.6%) were cohort/longitudinal studies, 7 (15.6%) were experimental research, and 3 (6.7%) were mixed-method studies (see Appendix A).

### 3.4. Summary of the Social-Emotional Domain Studied

The vast majority of the studies focused on the social-emotional (overall) development domain (*n* = 24, 53.3%), followed by the specific social competence (*n* = 7, 15.6%), emotional development (*n* = 5, 11.1%), social-emotional learning (*n* = 3, 6.7%), problem behavior (*n* = 3, 6.7%), self-regulation (*n* = 2, 4.4%), and social-emotional learning and problem behavior (*n* = 1, 2.2%) (see Appendix A).

### 3.5. Summary of the Key Findings of the 45 Included Studies

This systematic review study included studies on the social-emotional properties scale because both reviewers (GHY and JX) agreed that these studies are still relevant to one of the inclusion criteria, which is that the study focus should be on the social-emotional domain or either one of the social-emotional constructs. Moreover, there has been a lack of studies on the development and validation of social-emotional measures for young children, especially in Asian countries [9]. The studies on social-emotional properties included in this systematic review may help increase the understanding of the type of social-emotional measures validated in Asia and provide insights into the reliability and validity of the social-emotional measures used in Asian countries.

The reviewers (GHY and JX) read the full-text studies, and discussed and agreed on the six themes identified from the data extraction while considering specifically the purpose of this systematic review study. The reviewers (GHY and JX) ensured the coding consistency to review these 45 included studies systematically. The coding consistency was completed based on the social-emotional development domains studied. Under the emotional development theme, the reviewers reported the key findings of studies that examine children’s emotional development domain; the social-emotional learning and self-regulation themes were handled similarly. However, for the social-emotional development (overall), social competence, and problem behavior themes, the reviewers coded similar terms or definitions into one theme for simpler key findings. Some included studies in this systematic review may use different terms to describe and examine the same social-emotional domains. Under the social-emotional development (overall) theme, the reviewers reported the key findings for studies that do not specify a specific social-emotional domain construct. For the social competence theme, the reviewers collated studies that examined the social skills and social competence domains and reported their key findings. Thus, the six themes that were identified from the data that was extracted were: (1) social-emotional development (overall); (2) social competence; (3) emotional development; (4) problem behavior; (5) self-regulation; and (6) social-emotional learning. These six themes were then further divided into the names of the countries where the studies were conducted.

#### 3.5.1. Theme 1: Social-Emotional (Overall)

Social-emotional development (overall) was the most studied domain (*n* = 24, 53.3%) among the 45 studies reviewed. In China, social-emotional (overall) development in this systematic review referred to studies that do not specify a specific construct within social-emotional development. There are nine studies conducted in China that examined social-emotional (overall) development, of which five examined the association between social-emotional development and children’s academic skills [15,25,26].

The remaining three studies were on the family structure and home environment [50,52,53]. One study developed a social-emotional development scale for the Chinese population to measure boarding schools’ effect on left-behind children and their development [54].

Of the five studies that examined the association between social-emotional and children’s academic skills, most of the key findings were consistent with other studies, whereby there is a negative correlation between children’s social-emotional development and academic skills. Ren et al. [47] found that the father’s parenting does not mediate the outcomes of children’s social-emotional development and academic skills. This finding is inconsistent with the studies from other parts of the world, whereby parenting was reported to be one of the mediators in children’s social-emotional functioning and academic skills [72,73]. Ren et al. [47] reported that the lack of correlation with fathers’ parenting found in this study could be due to the self-reporting of idealistic parents rather than actual parenting practice. Many studies have reported gender biases in the parenting roles of children’s development within the Chinese culture [74,75]. Mothers are often the primary carer for the child, while fathers are less of the carer for the child and work toward providing for the family within the Asian culture [76]. Thus, it is possible that within the Chinese culture, Chinese fathers may be less involved in children’s development than Chinese mothers.

In Malaysia, there are two studies conducted on children’s social-emotional (overall) development. Mohamed et al. [42] conducted a cross-sectional study to examine the influence of family socioeconomic status on children’s social-emotional development. The researchers concluded that family socioeconomic status played a vital role in influencing children’s social-emotional development. Similar to Ren et al. [26], this study also reported no significant difference in children’s level of social-emotional development with the father’s income. However, there is a strong relationship between children’s level of social-emotional development and the mother’s education level and occupation (i.e., professional, semi-professional, non-professional, or unemployed) and the family’s income status. The findings from Ren et al. [47] and Mohamed et al. [42] further supported the suggestion that fathers might have a less significant parenting role within the Asian parenting culture that is the traditional parenting model of mothers as the primary carer for the family might still be dominant in Asia.

Mohamed et al. [43] conducted a cross-sectional study with 332 early childhood educators in Malaysia. This study found that the early educators in Malaysia had a suitable overview and knowledge of social-emotional development but needed knowledge of the factors that influenced the children’s development and how to foster it in the classroom. This study also reported a need to improve and equip teachers’ knowledge of social-emotional development. Goh et al. [37] developed and examined the feasibility of a Preschool Social-Emotional Competency Inventory to screen for children with poor competency. The Preschool Social-Emotional Competency Inventory (P-SECI) has a reliability score of 0.95 to 0.98 and proved valid in predicting children’s social-emotional competencies. Although this study reported high reliability scores, the researchers did not provide information on how the reliability and validity scores of the P-SECI were achieved. Furthermore, there was no information on whether the participant’s social-emotional competencies were measured before the pilot study. Thus, the reliability and validity of the P-SECI remain questionable as the information provided by the researchers was lacking.

In Singapore, Hamzah [38] conducted an interventional study that examined the efficacy of a speech and drama program on the social-emotional development of children with dyslexia [38]. The children who attended the intervention program showed improvement in their social-emotional functioning. However, this interventional study did not include a control group. Thus, this study might suffer from low internal validity. Moreover, participants in this study were aware of the purpose of the intervention program. Therefore, the results might have been affected by criterion contamination.

Ong et al. [45] conducted a 9-year longitudinal study on social-emotional development in Singapore to investigate whether parenting moderates between children’s early social-emotional competence and later mental health. The authors concluded that perceived parental care was associated with the early development of social-emotional functioning, while paternal care was especially important for children with more externalizing problems. In another study in Singapore, Yeo et al. [53] reported that children with physical disabilities were comparable in their self-esteem and academic achievements with typically developing children. However, children with physical disabilities experienced peer problems and were likely to participate less in school activities. This study’s findings were crucial as there has been scant research on inclusive education in the Asian context.

In Hong Kong, Li et al. [42] developed and validated a teacher-reported Chinese Inventory of Children’s Socioemotional Competence scale (CICSEC) for Chinese kindergarten children. The CICSEC was reported to have demonstrated suitable psychometric properties, whereby the overall CICSEC and the four subscales had excellent internal consistencies. The criterion validity analysis against the Strength and Difficulties Questionnaires (SDQ) [77] was positively correlated with school readiness and negatively with problem behavior. The CICSEC is one of the culturally relevant social-emotional scales developed in Asia, where the questionnaire items are consistent with the collectivistic culture in Asia. The exploratory and confirmatory analyses on the CICSEC have found that the children’s social-emotional competence was best represented by a four-factor model that included cognitive control, emotion expressivity, empathy and prosocial behaviors, and emotion regulations across child gender and grade level. This both exploratory and confirmatory finding of the four-factor model further confirmed (currently debated) view that children’s social, emotional, and cognitive skills are interdependent yet distinct [2,23,78]. Lam et al. [21] examined family characteristics associated with social-emotional development and found that boys had lower social-emotional competence than girls and that having more siblings positively enhanced social-emotional competence in children. Lam et al. [21] reported an interesting finding whereby boys from a non-Cantonese-speaking background were rated higher on the anger-aggression score than girls by their teachers who are mainly from a Cantonese-speaking background. Lam et al. [21] indicate that the differences found in the scores were possibly due to cultural differences in interpreting the student’s behavior. This finding is crucial as it indicates the need for more cultural studies to be conducted across Asia to understand social-emotional development in children. Another study by Lam et al. [41] was a Social-Emotional Well-Being of Early Childhood (SEWEC) interventional study that evaluated the social-emotional well-being of participants in an early childhood project in a Hong Kong kindergarten. The intervention program significantly improved social-emotional competence and reduced anxiety-withdrawal and anger-aggression in children post-intervention.

In Turkey, Bilir Seyhan et al. [47] examined the effects of the Promoting Alternative Thinking Strategies (PATHS) preschool program, which measured children’s social-emotional development, the perceived relationship between teachers and children, and the teachers’ ability to create positive classroom environments. This experimental study found that students in the intervention group significantly improved their social-emotional skills, interpersonal relationship skills, and emotion regulation compared to students in the control group. Furthermore, students and teachers in the intervention group perceived more positivity and dependency in their relationships than those in the control group. However, it is important to note that the observer in this experiment was not blinded to the study, and these findings might therefore be subject to observer bias. Another Turkish researcher performed a validation study on the Social-Emotion Assessment/Evaluation Measure-Preschool (SEAM) [36]. The confirmatory factor analysis for the Turkish SEAM supported the original factor structure. The researchers concluded that the SEAM measurement was a reliable and valid measure for the Turkish population.

In Thailand, Intusoma et al. [39] reported that educational television viewing benefits children’s social-emotional development. The study reported that 30 to 120 min of an educational program per day reduced the risk of poor social-emotional competence relative to non-viewers. However, television viewing for more than 2 h was reported to be unhealthy for social-emotional competence. Unfortunately, the researchers did not indicate whether the educational television program in this study was consistent across all participants. It is unclear which educational television program benefited the children’s social-emotional competencies. Van Driessche et al. [49] conducted a cross-sectional study in India to assess the predictive factors of caregiver burden and psychological comorbidities in families of children with hearing impairments. This study found that low educational attainment and domestic violence were associated with caregiver burden in parents of children with hearing disabilities. Dissatisfaction with family support, behavioral problems in children, and domestic violence were strong predictors for parental psychological morbidities, influencing social-emotional development in children with hearing disabilities. However, no formal hearing assessments were conducted on the participants involved in this study; thus, it is possible that children with mild hearing impairment were not recognized in this study and were considered to be of normal hearing.

In Korea, a cross-sectional study was conducted to examine the relationship between social-emotional development, gender, age, temperament, and maternal parenting behaviors [61]. Consistent with most studies, this study found that caregivers evaluated boys as having more externalizing behavior problems than girls. This study also found a negative correlation between children’s adaptability and behavior problems. Kim et al. reported that overprotective or permissive parenting is associated with low social ability in children [40]. On the other hand, refusal or neglect in parenting is associated with externalizing problem behaviors in children. In Indonesia, an observational study was conducted to examine the effect of an environmental education project approach on children’s social-emotional development [35]. This study found that the approach significantly improved children’s social-emotional development by 22% and increased children’s opportunities to interact with others.

#### 3.5.2. Theme 2: Social Competence

The social competence domain is the second social-emotional domain studied in Asia (*n* = 7, 15.6%). Five out of the seven studies on social competence used the term social competence, while the other two studies in Hong Kong used the term social skills to measure children’s social domain [58,79].

In Hong Kong, Ren et al. [79] conducted a longitudinal study to examine the antecedents, such as the child’s gender, family socioeconomic status, and extra-curricular activities involvement, associated with the children’s social skills. This study found that only children from a lower socioeconomic status who participated in extra-curricular activities had significant improvement in mathematics and reading skills and not those from a higher socioeconomic status. Moreover, this study also reported that parents of children from a higher socioeconomic status were more likely to enroll their children in non-academic extra-curricular activities than those of a lower socioeconomic status. This finding highlighted a possible connection between the socioeconomic background of parents and a preference for extra-curricular activities for their children (i.e., academic versus non-academic). However, Ren et al. [79] reported that there is no correlation between extra-curricular activities and the development of social skills.

Tong et al. [58] conducted a mixed-methods study to examine school-wide behavior interventions implemented in Hong Kong schools and explore teachers’ beliefs about the social skills programs implemented in schools. This study found that most teachers were aware of the benefit of the intervention and supported the school-wide behavior intervention implementation for children with social-emotional behavior difficulties. The findings in this study were consistent with the study in Malaysia, where teachers were aware of the benefit of social-emotional development. However, they lacked the training to implement the intervention in school [43]. The teacher’s knowledge of social-emotional development was not examined in the Hong Kong study.

In Korea and the United States, Lee et al. [56] conducted a cross-sectional study and found that both the Korean and United States students in the gifted group have a higher level of social competence than in the non-gifted groups. This study also found differences between Korean and United States gifted children in how they rated their social competence ability. The gifted Korean students rated themselves as better at resolving conflicts. In contrast, the gifted United States students rated themselves as better at asserting influence and getting along well with others. Furthermore, Lee et al. reported that female students were more able to make close friends than male students. This study’s finding contrasts with the other studies, which reported that gifted children could experience more difficulty creating relationships with their peers [80,81]. It is possible that the gifted students in this study could make friends with others because the samples were recruited from the same academic center so that the students may have the same level of giftedness. It is unclear whether gifted students can establish relationships with non-gifted students as easily.

In India, Bimla et al. [55] examined the association between social competency and children’s self-concept among rural children in India. This study found a significant positive relationship between children’s self-concept and social competence. Children who score higher on the self-concept scale were also found to have higher social competence scores on the Social Attributes Checklist. However, the findings from this study were not generalizable because the samples were too small. Moreover, this study did not elaborate on how the children were recruited from the schools. Yoleri [59] found a significant positive relationship between children’s temperament traits and social competence in Turkey. Yoleri also reported a significant positive relationship between the level of anger and the reactivity/withdrawal temperament. The findings from both studies were in line with past research from outside of Asia [82]. Thus, these findings suggested that children’s temperament and self-concept might predict their level of social competence despite cultural differences.

In Japan, Anme et al. [54] carried out a cohort study to describe the Interaction Rating Scale (IRS) features as an evidence-based index of children’s social skills and the quality of parenting. This study found that the IRS can measure children’s social skill development and the quality of parenting with high validity. However, there is no information on how the validity of the IRS was measured. Furthermore, there is also no demographic information on the clinical population in this study. This study could not be generalized as the validation for IRS used a cohort sample from the Japan Science and Technology Agency (JST) project. In Korea, Roh et al. [57] found that children who participated in a 7-week school-based social skill program significantly increased the quality of their peer relationships. Additionally, children from middle school who did not have peer relationships prior to the 7-week program were able to establish peer relationships with children of the same age. It is important to note that the effect of the school-based social skills program was measured using the Name Generator Question, where the participants were required to nominate the name of their peer that corresponded to each question, so a participant might nominate the name of a fellow peer without having established any relationship. Furthermore, this was a pre-post-intervention study. Hence, other factors might contribute to the result of this study.

#### 3.5.3. Theme 3: Emotional Development

Five studies in Asia studied the children’s emotional development domain (*n* = 5, 11.1%). Three out of the five studies on emotional development focused on maternal attitudes and children’s emotional development. All three studies reported that poor maternal attitude negatively correlated with children’s emotional development. 

In India, two studies reported that children’s self-reported emotional dysregulation partially mediated the relationship between the mother’s self-reported non-supportive attitude and children’s behavioral problems. Additionally, a factor analysis that compared maternal socialization behavior in the Asian context (India and China) found that parental expressive encouragement was unrelated to children’s problem behavior [64]. However, in a Western context, parental expressive encouragement was a supportive response. The findings of Raval et al. [64] were consistent with a study conducted in Korea [62]. The researchers in Korea reported that children with controlling maternal attitudes were negatively correlated with teachers’ reported behavior problems only [62]. Thus, these findings highlighted the normal differences in parental emotional expression and the importance of understanding children’s functioning in a cultural context.

In Pakistan, Akram et al. [60] compared the levels of adaptive emotional abilities between adolescents with hearing disabilities and normal-hearing adolescents. This study also examined the sociodemographic variables that might predict the emotional development of adolescents with hearing disabilities. Consistent with past studies, this study found that adolescents with hearing disabilities scored significantly lower on the adaptive emotional abilities scale than normal-hearing adolescents. Akram et al. [60] noted that hearing disability was not the only factor leading to poor adaptive emotional ability. The sociodemographic variables showed that the accessibility and availability of hearing and speech services, the presence of hearing-impaired family members, and the preferred communication language between the adolescent and family members were all associated with the adolescents’ adaptive emotional abilities.

In Taiwan, Chang et al. [61] carried out a study that assessed the level of emotional development among Taiwanese children based on Dabrowski’s theory [83]. The researchers also assessed whether emotional development and over-excitability predict personal adjustment in gifted and normal students. There were 123 mathematically gifted students and 132 normal students aged 16 to 18 years old involved in this study. Based on Dabrowski’s theory [83], emotional over-excitability is the most important characteristic that effectively predicts the level of emotional development among gifted students. However, this study found a negative correlation between emotional development and over-excitability. The researchers reported that cultural-specific variables might explain the discrepancy found in this study. Given the discrepancy in findings, it is important to note that this Taiwanese study only included mathematically gifted students, and the findings did not represent other gifted students in Taiwan.

#### 3.5.4. Theme 4: Social-Emotional Learning (SEL)

In China, Ye et al. [67] carried out a cross-sectional study with 375 early childhood educators to examine teachers’ perceptions of SEL in early childhood centers in China. This study found that teachers’ perceptions of SEL in China were at a moderate level. In addition, this study found that female teachers with both higher qualifications and teaching in a private school are more supportive of SEL than male or female teachers with lower qualifications or those teaching at public schools. These findings were consistent with the Malaysian study that examined teachers’ perceptions of social-emotional development in children [43]. However, Malaysian teachers had a lower understanding of the factors associated with social-emotional development in children, and the teachers reported that they did not know how social-emotional skills could be taught in a classroom setting.

In Thailand, Pinchumphonsan et al. [84] reported that a 15-week SEL intervention program significantly reduced the number of problem behaviors and increased positive behaviors in children who completed the program. However, similar to the other pre-post-intervention studies mentioned above, the improvement in participants’ behaviors could be attributed to other factors unrelated to the intervention program. Iaosanurak et al. [65] developed an SEL intervention program to compare Thai and Cambodian cultural groups and gender differences. However, the only significant finding was for gender differences in emotional intelligence, as Iaosanurak et al. found that the female participant’s empathy and responsibility scores increased after the intervention program while the male participant’s did not. Thailand and Cambodia are neighboring countries; the cultures in Thailand and Cambodia may be so similar that comparison across cultures led to a non-significant result.

In Hong Kong, Lam et al. [66] implemented and evaluated the effectiveness of a mindfulness program among four of the lowest academic tier groups in a public secondary school. This study found significant differences between the intervention and control groups with medium to large effect sizes on emotional control, working memory, self-monitoring, and anxiety/depression. The finding from Lam et al.’s study implied that the mindfulness program used, Learning to BREATHE (L2B), may be cross-culturally effective among the Asian population, and more studies in Asia are needed to validate the program further.

#### 3.5.5. Theme 5: Problem Behavior

Three studies among those reviewed studied children’s problem behavior (*n* = 3, 6.75%).

In China, Yang et al. [85] conducted a cross-sectional study to estimate the prevalence of behavioral problems and their risk factors among school children aged 6 to 16 years old in Beijing. There were 9295 students from urban and suburban districts in Beijing who participated in this study, and the detection rate of behavioral problems was 16.7%. However, the incidence of behavioral problems decreased with age. The researchers reported that behavioral problems were more significant in children aged 6–11 years and not significant in children aged 12–16 years. The researchers found that girls experienced more internalizing behavioral problems (e.g., depression, withdrawal, anxiety), and boys experienced more externalizing behavioral problems (e.g., aggressive behavior, social problems, hyperactivity). Yang et al. [85] also found that older children had a better level of social competence than younger children and theorized that the behavioral problems in the younger age groups might be due to China’s one-child policy. These children did not have siblings and might have had fewer opportunities to interact with other children outside school. In another study in China, Guo et al. [71] found prosocial behavior as a predictive factor for academic success and that peer acceptance mediated between prosocial behavior and academic success. These findings were consistent with the literature in the Western cultural context [86].

In Indonesia, Asri Dewi et al. [68] conducted a quasi-experimental study to assess the effect of playing a traditional game (Magoak-goakan) on developing prosocial behavior in preschool children in Bali. Fifty-two preschool children aged 5–6 years old were involved in this placebo control study (1:1 ratio). This study reported a significant increase in the intervention group’s prosocial behavior, while no changes were found in the control group. Although this is a small sample size study, it is worth noting that situational factors such as traditional game playing might influence the development of prosocial behavior. More research is required to examine the situational factors in children’s prosocial behavior in Asia.

#### 3.5.6. Theme 6: Self-Regulation

Two studies examined children’s self-regulation in Asia (*n* = 2, 4.4%).

Self-regulation is one of the domains in children’s development. Hot and cold self-regulation are the two aspects of self-regulation proposed by early researchers to avoid oversimplifying the self-regulation process and include the cognition and emotion domains [87]. Cool self-regulation requires the individual to solve emotionally neutral problems such as color or number sorting tasks. The cool self-regulatory task is similar to the executive function task. In contrast, hot self-regulation requires the individual to solve emotionally arousing problems, such as those involving delayed gratification. The hot self-regulatory task is often similar to the effortful control task [70,87].

In Hong Kong, Sun et al. [70] examined the development of cool and hot self-regulation in a Chinese sample of preschoolers in Hong Kong. This study also examined the relationships between cool and hot self-regulation and children’s academic achievement, behavioral problems, general knowledge, and fine motor and gross motor skills. Sun et al. [70] found that the cool self-regulation domain positively predicts children’s academic achievements, general knowledge, and fine motor and gross motor skills. In contrast, the hot self-regulation domain positively predicted children’s gross motor skills only. Both cool and hot self-regulation negatively predicted children’s hyperactivity.

Sun and Kang [70] also reported that cool and hot self-regulation might have a distinct structure among Chinese preschoolers. Sun and Kang could not find the exploratory factor analysis and confirmatory factor analysis fit that showed the relationship between cool and hot self-regulation that has been found in Western samples. The researchers concluded that Chinese culture differed from Western culture, whereby Chinese culture emphasized children’s behavioral conformity, obeying adults, and inhibiting inner impulses. The cultural differences found in this study might explain why most Chinese sample children in this study, as young as three years, could regulate their behaviors despite the emotional or motivational triggers in the hot self-regulation experimental task.

In China, Zhi et al. [71] explored the relationship between children’s self-control and family savings for children’s future education using the China Family Panel Studies data. This study found that children who lived in families with savings for their future education had higher self-control than families without savings. The effect size for this study was small (d = 0.06).

## 4. General Discussion

This systematic review aims to summarize the existing social-emotional research in Asia and provide suggestions for future research. In general, this systematic review found that although there are social-emotional development studies from various Asian countries, most of the studies were from East Asia (e.g., China and Hong Kong). There is a dearth of social-emotional development studies from other parts of Asia, such as Southeast, North, West, and Central Asia. The social-emotional behavior acceptable in a particular culture can vary across cultural groups and countries [25]. Most of the population in China and Hong Kong are of Chinese ethnicity, and these countries may have similar values for social-emotional competence. However, in other parts of Asia, acceptable social behavior or expression of emotions can be perceived differently by other cultural groups. For instance, Quah [88] reported that Malay and Indian parents were more likely to display affection toward their children than Chinese parents. Quah [88] also found that Chinese parents from lower socioeconomic status backgrounds were more inclined toward a parenting style in which the parent is always right, and children should be seen and not heard [89]. Culture plays a role in children’s social-emotional development and guidance from parents. Collie et al., Torrente et al., and Humphrey [90,91,92] have also called for more evidence-based practice and evaluations of the social-emotional development studies conducted in Asia, particularly among unique populations such as children with disabilities and ethnographic population groups. Therefore, more studies from different parts of Asia can provide a fuller picture of children’s social-emotional development throughout Asia.

Culture can affect parenting beliefs, values, and practices in raising children in a particular context [91]. A few studies (*n* = 10, 22%) in this review reported cultural differences in social-emotional development [25,45,46,52,61,62,63,65,70]. One study reported significant cultural differences in how children perceived social competence between Eastern and Western cultures [62]. Another study also reported that because of the heavy emphasis on behavioral conformity within Asian cultures, children in Asian cultures were reportedly able to regulate their behavior from as young as three years old [70]. These studies had their limitations in generalizing the results, albeit the significant findings in cultural differences. For example, the study by Sun et al. [70] only measured the cool and hot self-regulation domain among Hong Kong preschoolers. The extent to which the cultural differences found in this study occur in other Asian countries, where populations often have more than one ethnicity, remains unknown. The findings of this systematic review called for more cross-cultural comparison studies to better understand the possible cultural differences in children’s social-emotional development across Asia.

This systematic review also found that most social-emotional development studies in Asia focused on children’s academic achievement and family environment. These studies were mainly from China and Hong Kong and showed mixed findings on parenting and children’s academic achievement, especially between the urban and rural populations and the middle and lower socioeconomic status groups [54,76]. Ren et al. [79] reported that parents from lower socioeconomic status were more likely to enroll their children in academically related extra-curricular activities than in non-academically related activities. In contrast, Tan et al. [48] reported that caregivers from the rural population prioritized their children’s social-emotional development rather than academic achievement. Other studies have shown that children’s academic achievement might have been highly valued in Chinese cultures. Perhaps this is why many studies conducted in Asia are related to children’s academic achievement [69,93]. Nevertheless, there is still a lack of studies from other parts of Asia. Thus, it remains inconclusive that social-emotional development predicts academic achievement or that the family environment influences social-emotional development in all Asian cultures.

Gender differences in social-emotional competence have been reported in many international studies [94,95]. In many of the reviewed social-emotional development studies in Asia, consistent findings determined that boys had lower social-emotional competence than girls [22,40,85]. Both Kim et al. [40] and Yang et al. [85] reported that gender discrepancies in social-emotional development could be attributed to biological development in boys and girls. Both studies found that boys exhibited poor social behavior while girls exhibited emotional problems (i.e., depression and anxiety). Guo et al. [69] reported that the gender discrepancies in social-emotional development were dependent on the informant (i.e., parent, teacher, peers) and the social-emotional assessment used (i.e., rating, observation, nomination). This explanation is consistent with other studies where the parents were likely to report internalizing problems, and teachers were likely to report externalizing problems [85,96,97].

This systematic review also found that most of the intervention studies implemented as part of the various studies significantly improved children’s social-emotional development. However, most of these intervention studies were pre-post-intervention designs and did not have a control group [38,57,65,68]. There are possibilities for other confounding factors that might have been responsible for improving the social-emotional development post-intervention. In addition, only one or two intervention studies were conducted in each country, for example, Indonesia and Korea. Asri Dewi et al.’s [68] study consisted of only 52 samples, and Roh et al.’s [57] study used only the Name Generator Questionnaires to evaluate the effectiveness of the school-based social skills training program. Both studies have their limitations. The intervention’s effectiveness may not be generalized, as it has not been tested with a broader population.

Besides the paucity and uneven distribution of social-emotional development studies in Asia, this systematic review also found a need for high-quality studies. For instance, there was no information on the study samples except for the participants’ ages in Anme et al.’s study from Japan [54]. The study by Anme et al. [54] should have at least reported the participant’s gender distribution to prevent the premature conclusion that gender differences do not exist in any research [98]. Other information, such as the caregiver’s age and educational background, is also important as it could be a confounding variable in the research. Another study in this review also lacked information on how the samples were grouped into an intervention or control group [68]. The inadequate quality of the studies that have been conducted may lead to barriers to understanding children’s social-emotional development in Asia [99,100].

### 4.1. Research Gaps

This review identified several research gaps in the literature on children’s social-emotional development in Asia. First, there is a lack of ethnocultural studies and unevenly distributed studies across Asian countries. The Asian continent also includes other ethnicities besides Chinese culture. For instance, Singapore and Malaysia are multicultural countries that largely consist of Malay, Chinese, and Indian ethnicities. However, most social-emotional development studies in Asia were from East Asia, such as Hong Kong and China, where the culture is predominantly Chinese. The exceptions were from Japan and Korea. More studies are needed to provide an overall picture of social-emotional development in a multicultural country. Furthermore, the ethnocultural theory showed that cultural ethnicity might influence parental beliefs and behavior, guiding parenting practices and indirectly influencing children’s social-emotional development [89,101].

Secondly, there is a gap in the literature regarding whether the parental perception of social-emotional development differs between higher and lower socioeconomic status groups, as reported by Ren et al. [79]. The lower socioeconomic status group is reported to emphasize children’s academic achievement more than the higher socioeconomic group, which prioritizes children’s social-emotional development. As only one study explored this, the finding remained inconclusive. The differing view in parental perception of social-emotional development may be because of a lack of understanding of children’s social-emotional development among the lower socioeconomic status group. Given the importance of children’s social-emotional development and its long-term effect on children, more studies on parental perception of social-emotional development are needed. Parents’ perceptions can enhance or hinder children’s social-emotional development, so it is important to address this issue when examining social-emotional development across different cultures. There is an obvious need for more evidence-based studies in Asia, especially outside of East Asia. More quality studies in Asia may enrich the literature on children’s social-emotional development and provide directions for future research and improvements in approaches for prevention and intervention.

Lastly, three studies reported a lack of knowledge about children’s social-emotional development among teachers in Asia [43,58,67]. There is a need for more studies in Asia to understand teachers’ knowledge about social-emotional development. Besides parents, school teachers spend large amounts of time with children and are thus well-placed to identify problem behaviors early and to provide timely interventions for children at risk of poor social-emotional development [41]. However, early identification can only take place if teachers are well-equipped with the knowledge of social-emotional development and interventions. Thus, there is a need for interventional studies that improve and evaluate teachers’ knowledge of children’s social-emotional functioning in Asia.

### 4.2. Limitations

This review has several limitations. First, this review is limited to studies published only in English. There are potentially other relevant studies in other languages that may not have been included in this review. Second, this systematic review only included quantitative studies, which may have overlooked some studies in social-emotional development conducted using qualitative methods. Finally, there is a lack of information in some studies. Hence, the reviewers were unable to appraise some studies critically.

## 5. Conclusions

Despite these limitations, this systematic review contributes to the existing research on children’s social-emotional development in Asia by summarizing key knowledge areas and identifying critical gaps and directions for future research. It is important to note that most of the social-emotional development studies and theories are based on European and Western families [102]. Future studies should include more culturally diverse samples, especially from the Asian regions, so as to further explore the various aspects of social-emotional development within culturally relevant contexts.

Given that social-emotional development has gained increased attention from researchers, future studies should also consider collaboration between stakeholders to standardize the terms and definitions used to describe the domains of social-emotional functioning. Finally, parents and teachers play an important role in children’s social-emotional development [25]. Future research should consider filling the knowledge gap regarding parent and teacher perceptions of children’s social-emotional development in the Asian regions where there is a paucity of studies.

## Figures and Tables

**Figure 1 behavsci-13-00123-f001:**
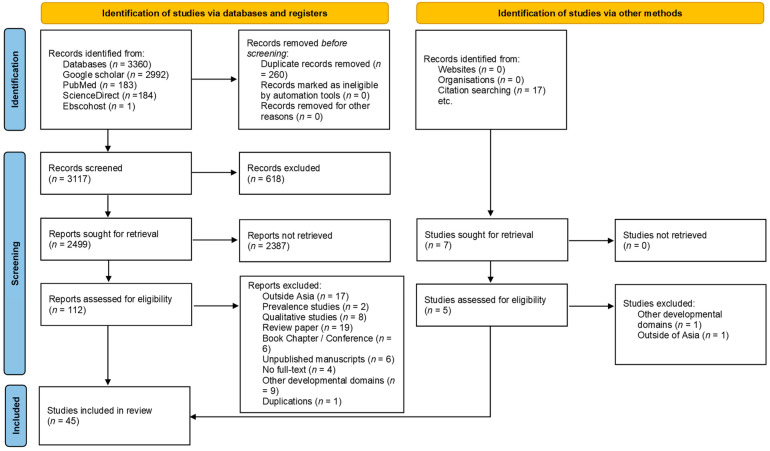
PRISMA flow diagram of the literature review on social-emotional development of children in Asia. Based on the Preferred Reporting Items for Systematic Reviews and Meta-Analyses framework [29].

**Table 1 behavsci-13-00123-t001:** A summary of included social-emotional development studies from Asia (*n* = 45).

Author, Year	Country	Age Range	Samples Description	Study Design	Social-Emotional Domains Studied	Summary of Key Findings
Abshor, U. 2017 [35]	Indonesia	3–4 years old	15 children (6 boys, 9 girls)	Cross-sectional	Social-emotional	A project conducted in early childhood environmental education setting that effectively raised children’s social-emotional development by 22% at the end of cycle-2.
Arslan Ciftci et al., 2019 [36]	Turkey	48–66 months old	394 children and their parent	Cross-sectional	Social-emotional	The Turkish Social-Emotional Assessment/Evaluation Measure (SEAM) showed suitable linguistic equivalence, validity, and reliability.
Goh et al., 2019 [37]	Malaysia	5–6 years old	49 students, their parent, and teacher	Cross-sectional	Social-emotional	Preschool Social-Emotional Competency Inventory (P-SECI) showed high reliability index of 0.98 for Teachers and 0.95 for Parents.
Hamzah, M. 2019 [38]	Singapore	7–11 years old	6 dyslexia students and their parent, and 2 teachers	Mixed- method (Quantitative and Qualitative)	Social-emotional	Children with dyslexia showed 20.5% improvement in the Southampton Emotional Literacy Scales (SELS) scores after attending the Speech and Drama Arts (SDA) program for one year.
Intusoma et al., 2013 [39]	Thailand	1 and 3 years old	4157 children	Cohort study	Social-emotional	Viewing of 20–30 min/day was associated with a decreased risk of low social-emotional competence (SEC) compared to non-viewers after adjustments for confounding factors.
Kim et al., 2011 [40]	Korea	1–2 years old	51 infants and their parent (30 boys 21 girls)	Cross-sectional	Social-emotional	The adaptability of infants showed a negative correlation with externalizing problem behaviors. The boys’ social competence scores were significantly lower than the girls’ scores when controlled for similar age and gender.
Lam et al., 2016 [21]	Hong Kong	3–6 years old	1326 children and 106 early child educators	Cross-sectional	Social-emotional	The anger-aggression scores of boys from a non-Cantonese-speaking background were higher than girls, rated by their Cantonese-speaking teachers.
Lam et al., 2017 [41]	Hong Kong	3–6 years old	990 children (87 clinical diagnoses of autism, ADHD, Asperger’s, dyslexia, and intellectual disabilities) and 106 teachers	2-months Intervention (Experimental research)	Social-emotional	The Social-Emotional Well-Being of Early Childhood (SEWEC) Intervention Project developed based on the Wisconsin Pyramid Model significantly improved social competence and reduced anxiety-withdrawal and anger-aggression in kindergarten children aged 2.5–6 years old.
Li et al., 2020 [42]	Hong Kong	3–6 years old	1731 children	Cross-sectional	Social-emotional	Chinese Inventory of Children’s Socioemotional Competence (CICSEC) demonstrated excellent internal consistencies. The criterion validity was positively correlated with school readiness (rs ranging from 0.32 to 0.68) and negatively with problem behaviors (rs ranging from −0.27–0.07).
Mohamed et al., 2020 [43]	Malaysia		332 early childhood educators	Cross-sectional	Social-emotional	Malaysian early childhood educators have a moderate perception of social-emotional development, and demonstrated a poor understanding of the factors associated with social-emotional development and how social-emotional strengths can be taught in the classroom.
Mohamed et al., 2018 [44]	Malaysia	3–4 years old	237 children	Cross-sectional	Social-emotional	Children’s level of social-emotional development was closely associated with mother’s education level, mother’s occupation, and father’s income, showed an average relationship to the father’s education level, and a poor relationship with the father’s occupation.
Ong et al., 2017 [45]	Singapore	7–9 years old	445 children and their parent	Longitudinal (9 years)	Social-emotional	Perceived parental care was found associated with the quality of socioemotional development, while optimal parenting by the father was essential for children with more externalizing problems in childhood.
Ren et al., 2016 [7]	China	3–5 years old	154 parents of preschool children	Mixed-method (quantitative and qualitative)	Social-emotional	Parents placed more importance on children’s social-emotional skills and compliance than academic skills.
Ren et al., 2020 [26]	China	5–6 years old	336 Chinese children and their parents	Longitudinal (7 months)	Social-emotional	Relation between co-parenting quality and children’s academic readiness was mediated by children’s behavioral regulation, except for the father’s parenting practices.
Ren et al., 2016 [25]	China	3–6 years old	154 parents (133 mothers 21 fathers)	Cross-sectional	Social-emotional	Children’s withdrawn behaviors and attention problems were negatively related to their preacademic skills. Parent- and teacher-reported positive social behaviors were positively related to children’s preacademic skills.
Ren et al., 2020 [46]	China	3–6 years old	695 preschoolers and their parent	Longitudinal (1 year)	Social-emotional	Extra-curricular involvement was positively associated with children’s cognitive and language development, but not with social-emotional development, after controlling for demographic variables and children’s prior performance.
Bilir Seyhan et al., 2017 [47]	Turkey	4–6 years old	560 students and 41 teachers	Experimental study	Social-emotional	The Promoting Alternative Thinking Strategies (PATHS) Intervention group (IG) teachers reported more improvement in children’s social-emotional skills, interpersonal relationship skills, and emotion regulation. IG children showed a higher level of prosocial behavior, increased compliance, better problem-solving skills, and more positive feelings.
Tan et al., 2020 [48]	China	0–3 years old	847 left-behind children (either one or both parents have migrated for work)	Cross-sectional	Social-emotional	37.2% of left-behind children had social-emotional problems, and 40% of caregivers reported depressive symptoms. Caregiver depressive symptoms positively correlated with social-emotional problems in left-behind children, and the mediation by the home environment was 15.6% of the total effect.
Van Driessche et al., 2014 [49]	India	3–16 years old	201 parents/caregivers of children with hearing impairment and 104 parents/caregivers of normal-hearing children	Cross-sectional	Social-emotional	Low educational attainment and domestic violence were associated with caregiving strain.
Wang et al., 2020 [50]	China	6–24 months	1809 infants	Cross-sectional	Social-emotional	54.0% of children were at risk of developmental delay, 60.3% at risk of language delay, 36.3% at risk of motor delay, and 40.6% at risk of the social-emotional problem. Quality of the family environment was significantly associated with the child’s development.
Wang et al., 2019 [51]	China	10–13 years old	975 students of single-parent and two-parent families (431 single-parent and 544 two-parent)	Cross-sectional	Social-emotional	Children from two-parent families scored significantly higher on measures of social-emotional development than single-parent families.
Wang et al., 2020 [14]	China	9–14 years old	6638 boarding school students	Cross-sectional	Social-emotional	A break every 2–3 weeks had positive impacts on boarding school students, while every four weeks or more had negative effects on boarding school students. Taking a break every 2–3 weeks had a more positive effect on both left-behind children and commuting daily between home and school students.
Wang et al., 2017 [52]	China	9–14 years old	6638 boarding school students	Cross-sectional	Social-emotional	Left-behind children’s social-emotional competence was significantly lower than those under parental guardianship. Left-behind children living on campus had a higher negative social-emotional competence than left-behind children that commute daily.
Yeo et al., 2018 [53]	Singapore	8–16 years old	60 children (30 physical disability and 30 typical development)	Cross-sectional	Social-emotional	Children with physical disabilities met academic expectations in school and had comparable self-esteem but experienced peer problems and participated less in school activities.
Anme et al., 2010 [54]	Japan	18–42 months, 7 years old	823 children and their caregivers	Cohort study	Social competence	Interaction Rating Scale (IRS) is a reliable, valid, feasible, and practical tool.
Bimla et al., 2012 [55]	India	9–13 years old	44 children	Cross-sectional	Social competence	There was a significant increase in social competence with self-concept.
Lee et al., 2012 [56]	U.S. and Korea	12–17 years old	740 gifted students (373 U.S., 367 Korea)	Cross-sectional	Social Competence	Gifted students positively perceived their interpersonal ability and peer relationships at a level comparable to or higher than non-gifted students. Female students in both the Korean and American samples were reportedly more positive in rating their profiles of interpersonal ability and peer relationships compared to male students.
Roh et al., 2018 [57]	Korea	10–12 years old	90 students	7-weeks intervention (experimental study)	Social skills	The social skills training program significantly increased peer relations.
Tong et al., 2012 [58]	Hong Kong		60 teachers	Mixed-method (quantitative and qualitative)	Social competence	Most teachers believed that behavioral and social skill programs should be implemented in schools at an early stage. Teacher’s professional development in social skills training, teacher’s belief and attitude, and the contextual support within the school for the school-wide intervention was found to influence the effectiveness of the school-wide interventions.
Yoleri, S. 2014 [59]	Turkey	5–6 years old	112 children, their mothers, and teachers	Cross-sectional	Social competence	Social competence level had significant positive relationship with the persistence and rhythmicity level of temperament traits. A significant positive relationship was between the level of anger-aggression and the reactivity temperament trait subscales on the Social Competence and Behavior Evaluation Scale for Children (STSC). Social competence had a significant relationship with temperament traits.
Akram et al., 2014 [60]	Pakistan	12–18 years old	469 hearing impairment students and 1050 normal-hearing students	Cross-sectional	Emotional development	The Adaptive Emotional Abilities Scale (AEAS) was found to have acceptable face and content validity, internal and test-retest reliability. Participants with normal hearing scored significantly higher on the AEAS than participants with hearing impairment.
Chang et al., 2019 [61]	Taiwan	15–18 years old	255 gifted students (123 mathematically gifted and 132 regular students)	Cross-sectional	Emotional development	Gifted students had better emotional adjustment than the normal students. Social-emotional development positively correlated with the intellectual over-excitability, but were negatively correlated with over-excitability (EOE). Intensive emotional over-excitability (EOE) significantly predicted personal maladjustment.
Lee et al., 2017 [62]	Korea	4–6 years old	70 preschoolers, their mother, and teachers	Cross-sectional	Emotional development	Children’s emotional understanding was negatively correlated with teacher-reported behavior problems and positively associated with social competence. Controlling maternal attitude toward children’s positive emotional expressions was negatively correlated only with teacher-reported behavior problems. Maternal attitude toward children’s positive emotional expressiveness moderated the relationship between emotional understanding ability, and behavior problems and social competence.
Raval et al., 2014 [63]	India	11–12 years old	110 mothers and their children	Cross-sectional	Emotional development	Suburban Indian mothers were more likely to endorse relational socialization goals than autonomous socialization goals. Children’s self-reported dysregulation partially mediated the positive association between the report of the mother’s non-supportive behaviors and child behavior problems.
Raval et al., 2018 [64]	India and China	10–12 years old	305 mothers and their children (147 India 158 China)	Cross-sectional	Emotional development	Mothers’ supportive responses and child emotional regulation sequentially mediated maternal relational socialization goals and child internalizing problems. Children’s emotion dysregulation mediated the relation between maternal non-supportive responses and child externalizing problems.
Iaosanurak et al., 2015 [65]	Thailand	11–12 years old	23 children	8-weeks intervention (experimental study)	Social-emotional learning	No significant difference in social-emotional learning competencies between the Thailand and Cambodia students. Only female students in both countries showed a significant increase in empathy and responsibility at post-intervention.
Lam et al., 2020 [66]	Hong Kong	11–15 years old	115 students	5-months intervention (experimental study)	Social-emotional learning	Improvement in the L2B group and deterioration in the control group (IAU) was observed on emotional control, working memory, self-monitoring and anxiety/depression.
Ye et al., 2020 [67]	China		375 teachers	Cross-sectional	Social-emotional learning	There were significant differences in teachers’ perceptions of social-emotional learning based on teacher qualification and the type of school. No significant differences in school location and teaching experience.
Asri Dewi et al., 2018 [68]	Turkey	5–6 years old	52 children	Experimental study	Prosocial behavior	Traditional game Magoak-goakan has a positive influence on the development of prosocial behavior in the intervention group.
Guo et al., 2018 [69]	China	11–12 years old	456 students	Cross-sectional	Prosocial behavior	Children’s prosocial behavior positively predicted their academic achievement, and peer acceptance played a mediating role in the pathway.
Sun et al., 2020 [70]	Hong Kong	3–5 years old	951 children and their mothers	Cross-sectional	Self-regulation	Cool self-regulation was found to predict children’s achievement differently. Cool self-regulation was found to predict children’s early academic learning, general knowledge, and fine and gross motor skills. Hot self-regulation only positively predicted children’s gross motor skills. Both cool and hot self-regulation were found to negatively predict children’s hyperactivity level.
Zhi et al., 2020 [71]	China	10–15 years old	2182 children and their parent	Cohort study	Self-regulation	Family savings for children were positively associated with children’s level of self-control.

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
