# Peer review of "Social-Emotional Development of Children in Asia: A Systematic Review"

_behavsci, 2023, doi:10.3390/bs13020123_

Round 1

Reviewer 1 Report (Previous Reviewer 2)

Thank you for the opportunity to re-review manuscript “behavsci-2058693”.

The authors have responded to my comments 1-4 and 6-9. The response to my initial concern 5 is partly ok, partly not comprehensible. E.g., ll. 266-267, it does not become evident what the authors are trying to say. The authors suggest that the validation studies (Which findings of which validation study?) also provide “an answer” to an “aim” of the current review, but but fail to provide an intelligible explanation. Also, the rest of the paragraph contains incorrect use of the English language.

I included a few minor mainly grammar or spelling errors in the following, among many more in the manuscript:

- ll. 78: Sentence starting with “Children with authoritarian parenting…” à There are a couple of mistakes in this sentence, should be, e.g., “higher internalizing problems” without the “a”.

- ll. 82: “Based on the Western cultures’ definition of social competence, children from authoritarian parenting may have (…)” à this is also difficult to understand: Children cannot be “from” authoritarian parenting. Suggestions: children exposed to authoritarian parenting?

- ll. 171-3: This sentence is also not correct, suggestion: “The STROBE statement assessment tool is a 22-item checklist (…)”. The rest should be corrected by the authors, pls.

- l. 197: The authors are summarizing the studies according to the country of each study, correct? Then it should be “the study’s country”, not the “country’s study”.

- in Table 1, description of the study of Lam et al.: Sentence should probably be “The boys’ social competence scores were found significantly lower compared to the girls’ scores when controlled for similar age and gender.”? Second sentence should probably be “Boys from a non-Cantonese speaking background scores higher on the anger-aggression score than girls rated by their Cantonese speaking teachers.” ?

To summarize, most of my comments have been satisfactorily dealt with. Unfortunately, the revised paragraphs lack care in terms of grammar and sentence structure. New errors are introduced in the revised sections. I strongly recommend to make use of a professional English language editing before resubmission. In its current version, the paper is not suitable for publication.

Author Response

Reviewer 2 Report (Previous Reviewer 3)

Dear Authors,

The manuscript looks better now.

I wish you good luck with your further research.

Author Response

Reviewer 3 Report (New Reviewer)

Dear Authors, 

First of all, I would like to acknowledge the effort you have made to carry out this review. Here are some recommendations that I believe may help to improve the paper. 

First of all, I recommend a thorough revision of the writing, the paper would benefit from it. 

In addition, I would recommend specifying the inclusion and exclusion criteria in greater detail, as sometimes they include criteria that are inconsistent with each other, and on other occasions they include information that could perhaps be included in the theoretical framework. I attach a particularly problematic excerpt from my point of view: 

"This systematic review study will include studies where social-emotional 140

functioning is not the main focus of the study but are related to social-emotional domains 141

to have an overview of the extent and depth of social-emotional development research 142

conducted in Asia. The definition of children varies across Asia. Thus, this review study 143

defined children as any person aged 0 - 18 years old. This definition is consistent with the 144

UNICEF Convention on the Rights of the Child definition [29]. Asia is the world's largest 145

and most populous continent that constituted about 60% of the world's population [30]. 146

The Asia continent consists of six main geographical regions; Northern Asia, Western 147

Asia, Central Asia, Eastern Asia, Southern Asia, and Southeast Asia "

On the other hand, Figure 1 looks blurry. 

In addition, the length of Table 1 makes it difficult to read the paper. 

In addition, since you explain that a coding process has been followed, I invite you to explain this process better. 

Best wishes, 

Reviewer

Round 2

Reviewer 3 Report (New Reviewer)

My concerns have been meet. 
Best regards, 

Reviewer

This manuscript is a resubmission of an earlier submission. The following is a list of the peer review reports and author responses from that submission.

Round 1

Reviewer 2 Report

Thank you for the opportunity to review manuscript “behavsci-2058693”. The study addresses an important and current topic and aimed to systematically review the current knowledge and studies on socio-emotional development of children in Asia. The systematic review is in general methodologically sound and well-written. I have a few suggestions on how to improve the manuscript in order to make it suitable for publication in Behavioural Sciences.

Major:

1.      The authors provide a nice overview in the introduction section, but should provide more details on the theoretical models of social-emotional development (e.g., which they cite already cite like Bronfenbrenner’s model, ll. 32-34).

2.      The paragraph on cultural differences is vital to the paper (ll. 49-59), and also needs some more theoretical background. The authors should provide more references.

3.      The Methods section is very well-written and comprehensible.

4.      In the Results’ section, sometimes, presentation of data in a figure and the text seem redundant, e.g., the presentation of data in figure would suffice. Figure 4 and Figure 5 seem unnecessary as the data is presented in a better way in paragraph 3.3 and 3.4 respectively. The authors should change that.

5.      Results’ section, table 1: In table 1, summaries of the key findings of the studies are provided, but the authors sometimes use abbreviations, which were not introduced (e.g., SEC). Also, the studies by Arslan Cifti et al., 2019, Akram et al., 2014, Name et al., 2010, Li et al., 2020 and Goh et al., 2019 (among others) seem to be mainly studies on the properties of a certain scale in an Asian country. The authors should provide a statement, why these studies were included in the review. Also, the summary of Lam et al., 2016 is unintelligible and should be revised. The summary of Bimla et al., 2012 does not provide enough detail for the reader to understand, what the study investigated. Also, in some of the summaries, there are spelling or grammar errors, which the authors should correct for readability purposes.

6.      Results’ section, 3.5.1: This section “Social-Emotional (Overall)” is very long and lacks clear structure and focus. The manuscript would profit greatly from more thematically definable subheadings/paragraphs.

7.      The Results’ section should be revised with respect to the English language, see also some corrections below.

8.      In some parts of the Results’ section, terms such as, e.g. “cool” and “hot” self-regulation are used without further explanation. The authors should provide some background on mentioned concepts throughout the Results’ section.

9.      In the Discussion section, the authors provide gaps of the literature, which is a readable section. They should also report on the lack of cross-cultural comparative studies.

Minor:

In the following, I point out some of the grammar/spelling errors, which I detected, but this list is probably incomplete.

Abstract:

- l. 21, should be “Asian”.

Introduction:

- l. 43, should be “that lead to success in school, work, and daily life [9].”

- l. 60, sentence should not begin with “As mentioned”.

- l. 72, should be “Which Asian countries have conducted studies on […]?”

Materials & Methods:

- l. 81, should be “research”

- l. 116, should be “ to reduce the definition bias of “Asia”” which “Asia” in quotation marks.

- l. 139 should be “each study’s findings was entered”

- l. 150, should be “Following the screening of the titles and abstracts of the studies”

- l. 154, should be “by the country’s study”

Results:

- l. 170, should be “parents and teachers, and 689 were children with disabilities”

- l. 254, should be “similar to the study of Ren et al.”

- l. 305, should be “emotion regulation”

- l. 326, should be either “Other Turkish researchers” or “Another Turkish researcher”

- l. 356, should be “refusal or neglect in parenting” or “refusing and neglectful parenting”

- l. 272, should be “also reported that parents of children”

- l. 390, should be “students in the gifted group had a higher level”

Discussion:

- l. 560, should probably be “inclined towards a parenting style, in which the parent is always right”?

Reviewer 3 Report

Dear authors,

You will find my comments and suggestions in the file attached.
